# The Interplay between Radio AGN Activity and Their Host Galaxies

Guilherme S. Couto [1],[†] and Thaisa Storchi-Bergmann [2],[*],[†]

1 Leibniz-Institut für Astrophysik Potsdam, An der Sternwarte 16, 14482 Potsdam, Germany; gcouto@aip.de
2 Instituto de Fisica, Universidade Federal do Rio Grande do Sul, CP 15051, Porto Alegre 91501-970, RS, Brazil
* Correspondence: thaisa@ufrgs.br
† These authors contributed equally to this work.

**Abstract:** Radio activity in AGN (Active Galactic Nuclei) produce feedback on the host galaxy via the impact of the relativistic jets on the circumnuclear gas. Although radio jets can reach up to several times the optical radius of the host galaxy, in this review we focus on the observation of the feedback deposited locally in the central region of the host galaxies, in the form of outflows due to the jet-gas interaction. We begin by discussing how galaxy mergers and interactions are the most favored scenario for triggering radio AGN after gas accretion to the nuclear supermassive black hole and star formation enhancement in the nuclear region, observed in particular in the most luminous sources. We then discuss observational signatures of the process of jet-gas coupling, in particular the resulting outflows and their effects on the host galaxy. These include the presence of shock signatures and the detection of outflows not only along the radio jet but perpendicular to it in many sources. Although most of the studies are done via the observation of ionized gas, molecular gas is also being increasingly observed in outflow, contributing to the bulk of the mass outflow rate. Even though most radio sources present outflow kinetic powers that do not reach 1% $L_{bol}$, and thus do not seem to provide an immediate impact on the host galaxy, they act to heat the ISM gas, preventing star formation, slowing the galaxy mass build-up process and limiting the stellar mass growth, in a "maintenance mode" feedback.

**Keywords:** active galaxies; galaxy evolution; galaxy jets; galaxy kinematics and dynamics; ISM





## 1. Introduction

The Active Galactic Nuclei (AGN) population can be represented by two main categories. In the first category, Quasars and Seyfert galaxies, sources of high bolometric luminosities, capable of generating winds through radiation pressure due to its accretion rate close to Eddington, usually found in wide angle outflows, are commonly distinguished as *radiative*-mode (or sometimes also called as *quasar*-mode) AGN. In the second category, the so called *radio*-mode AGN (or *jet*-mode, or *kinetic*-mode), the central engine launches powerful collimated jets of relativistic particles accelerated in the inner regions of the accretion disk due to its intense magnetic fields. The origin of the difference between these two categories is believed to happen within the accretion disk structure and internal thermodynamics, and the resulting mass accretion rates [1] (and references therein).

Radio emission is one of the most distinctive tracers of nuclear activity in galaxies. The radio jets can be extremely powerful, extending up to Mpc scales and producing strong feedback in the surrounding medium of early-type galaxy hosts in the center of galaxy clusters [2–5]. But do radio jets produce feedback in its host galaxies? This is the central topic of this short review of the interplay between radio-emission from AGN and its host galaxies. Fast ($\gtrsim$1000 km s$^{-1}$) H I 21 cm absorption outflows observed using WSRT [6,7] and X-ray detections related to shocks signatures due to the jet-gas interactions, e.g., observed with Chandra [8,9] illustrate how multi-wavelength analysis is fundamental to properly characterize the role of radio feedback.

With the emergence of Integral Field Spectroscopy (hereafter IFS) optical instruments in the past years, such as the Gemini GMOS [10], VLT MUSE [11] and GTC MEGARA [12], along with deep observations in other wavelength instruments such as ALMA [13] and most recently JWST, recent studies of outflows in local AGN have been able to characterize and constrains the outflow properties, resolving their kinematics and determining its extent within the host galaxies. High resolution spectra allow to extract information in the ionized and molecular gas phases, such as velocity dispersion and emission line ratios, helping understand how gas excitation works. This has been proven useful in radio-loud AGN, characterized by shock-driven outflows due to jet-gas interactions.

We begin by discussing the role of interactions in the triggering of radio activity in galaxies in Section 2, since there is plenty of evidence that the host galaxies of the most powerful radio galaxies show morphological disturbances characteristic of interactions e.g., [14]. In Section 3 we discuss the signatures of radio-mode feedback within galaxies, as evidenced by the disturbed ionized gas kinematics in association with radio sources e.g., [15]. This association is most frequently observed in the optical, in particular in recent studies IFS of the host galaxies [16,17]. These signatures point to an important role of shocks associated to the radio jets in producing feedback in the host galaxies.

A recent development in the study of AGN feedback is the observation of enhanced velocity dispersion perpendicular to the ionization axis and radio jet, and this is discussed in Section 4 e.g., [18,19]. In Section 5 we discuss how the increasing sensitivity of radio antennas is allowing us to find radio jets down to very low powers, as observed in the so-called "Red Geysers" [20]. These galaxies show a faint AGN with mild ionized gas winds but that extend to large distances from the nucleus. The frequent presence of a radio source at the nucleus [21], suggests an association of the radio source with the winds, that seem to provide a "maintenance mode" feedback. In this kind of feedback, the impact of the outflows is mild, but enough to heat and disturb the surrounding gas precluding the formation of new stars, as also been argued to happen in the central galaxies of clusters, halting the so-called "cooling flows" e.g., [2].

## 2. Galaxy Interactions as Triggers of Radio AGN Activity

We begin the discussion of radio activity in galaxies via its triggering, and one of the possible mechanisms are galaxy interactions and mergers. While the occurrence of minor mergers seems to be frequent in galaxies hosting low-luminosity AGN ($L_{AGN} \leq 10^{44}$ erg s$^{-1}$), major mergers are dominant in the AGN high-mass regime (SMBH masses $\geq 10^8$ M$_\odot$) [22–24]. For the case of AGN radio activity, these sources are usually related to early-type hosts, what makes it easier to find morphological disturbances in images, non-orbital motions in the kinematics, and support a connection with galaxy interactions as they are found in the center of galaxy clusters [25–27].

Galaxy interactions as a mechanism to produce dynamical destabilization of large amounts of gas have been considered a possible trigger of the AGN feeding, driving the gas from kpc-scale distances to the nucleus of the galaxy [28,29]. In a broad scenario, a major merger of gas-rich progenitors could be the responsible to concentrate gas in the central regions of the resulting galaxy, possibly triggering a starburst event close to the nucleus, while part of the gas makes its way to the central SMBH, initiating AGN activity. This scenario has been proposed in a number of powerful AGN, suggesting a link between galaxy mergers and both starburst and AGN activities [30–33].

Using Gemini GMOS broad-band images of a sample of 46 2 Jy radio galaxies, Ramos Almeida et al. [14] performed one of the first qualitative studies of host galaxy morphologies of radio-loud AGNs at redshifts in the range $0.05 < z < 0.7$. Peculiar morphologies including fan-like structures, tails, bridges, shells and others were observed in 85% of the sample galaxies, a considerably higher fraction than observed in quiescent elliptical galaxies [34,35]. When dividing the sample into Weak- and Strong-Line Radio Galaxies (WLRGs and SLRGs, respectively, with WLRGs presenting [O III]$\lambda$5007Å emission line equivalent width below 10Å), the authors find an incidence of disturbed morphologies

in 94% of the SLRGs, while WLRGs present merger signatures in only 27% of the galaxies, indicating there is a correlation between the AGN power and the incidence of mergers.

More recent studies are in agreement with these results [36]. With deep imaging observations of 30 intermediate radio power AGN, Pierce et al. [37] have found that the most radio-powerful half of the sample displays higher incidence of interaction signatures than the less-powerful half ($67 \pm 12\%$ and $40 \pm 13\%$, respectively, with corresponding radio luminosities of $23.06 < \log(L_{1.4\,GHz}) < 24.0$ W Hz$^{-1}$ and $22.5 < \log(L_{1.4\,GHz}) < 23.06$ W Hz$^{-1}$). Even though there is no clear correlation between $L_{1.4\,GHz}$ and $L_{[O\,III]}$ luminosities, the same fractions are observed when separating the sample in terms of $L_{[O\,III]}$.

In this context, Ultra Luminous Infrared Galaxies (ULIRGs) are also important as they support the scenario of luminous nuclear starbursts triggered by merging galaxies that evolve to become luminous AGN (as will be further explored in the next section). As shown by Sanders and Mirabel [38] using IRAS observations, the properties of ULIRGS indicate that the incidence of mergers correlate with that of AGN at the highest IR luminosities (see their Table 3). The mean projected separation of the nuclei also decreases with the increase of AGN and merger fraction, suggesting that the AGN activity is more likely to happen at latter phases of the galaxy merger due to gas migration to the central regions, which also feeds the very high star formation rates observed in ULIRGs of up to $\sim$1000 M$_\odot$ yr$^{-1}$.

*Individual Studies*

Clear signatures of galaxy interactions in radio-loud AGN have been largely observed in individual studies. This is the case of B2 0648+27, an early-type galaxy presenting interacting signatures such as tail-like structures in low-surface brightness emission [39]. Emonts et al. [30] found three main events in the galaxy history while studying its neutral gas, optical spectroscopy and radio continuum: a major merger that happened $\gtrsim$1.5 Gyr ago; a starburst $\sim$0.3 Gyr ago; and the AGN activity that must have started $\gtrsim$0.001 Gyr ago. The starburst event was succeeded by the AGN activity after the removal of angular momentum from the gas to be driven towards the nucleus. B2 0648+27 seems to be between a starburst and a normal elliptical galaxy in an evolutionary sequence triggered by galaxy interaction and where the AGN activity happens in between.

Another interesting study case is the host galaxy of the radio-loud AGN 4C +29.30. This galaxy seems to be a merger system in its late stages, with a characteristic dust lane passing in front of the central region, in a similar fashion to Centaurus A, a well studied radio galaxy with clear signatures of past interactions [40,41]. As in Centaurus A, 4C +29.30 also presents several phases of recurrent radio activity, with structures corresponding to different time periods, from $\gtrsim 200$ Myr to a very young activity of only $10^4$ yr [42,43], which could relate to different feeding events during the merger process. As shown in Figure 1, 4C +29.30 shows a very complex morphology not only in the optical, but also related to the jet in X-rays and radio wavelengths. As we discussed in a paper detailing Gemini-GMOS IFU data [16], powerful ionized gas outflows are being launched due to the interaction between the relativistic jet and the circumnuclear gas, a scenario quite common in radio-loud AGN which we will explore in the following sections. Several other studies on individual sources display a possible relation between radio AGN activity and galaxy interactions, which may be related ultimately to the fueling of the SMBH, or at least driving gas to the nuclear vicinity e.g., [44–46].

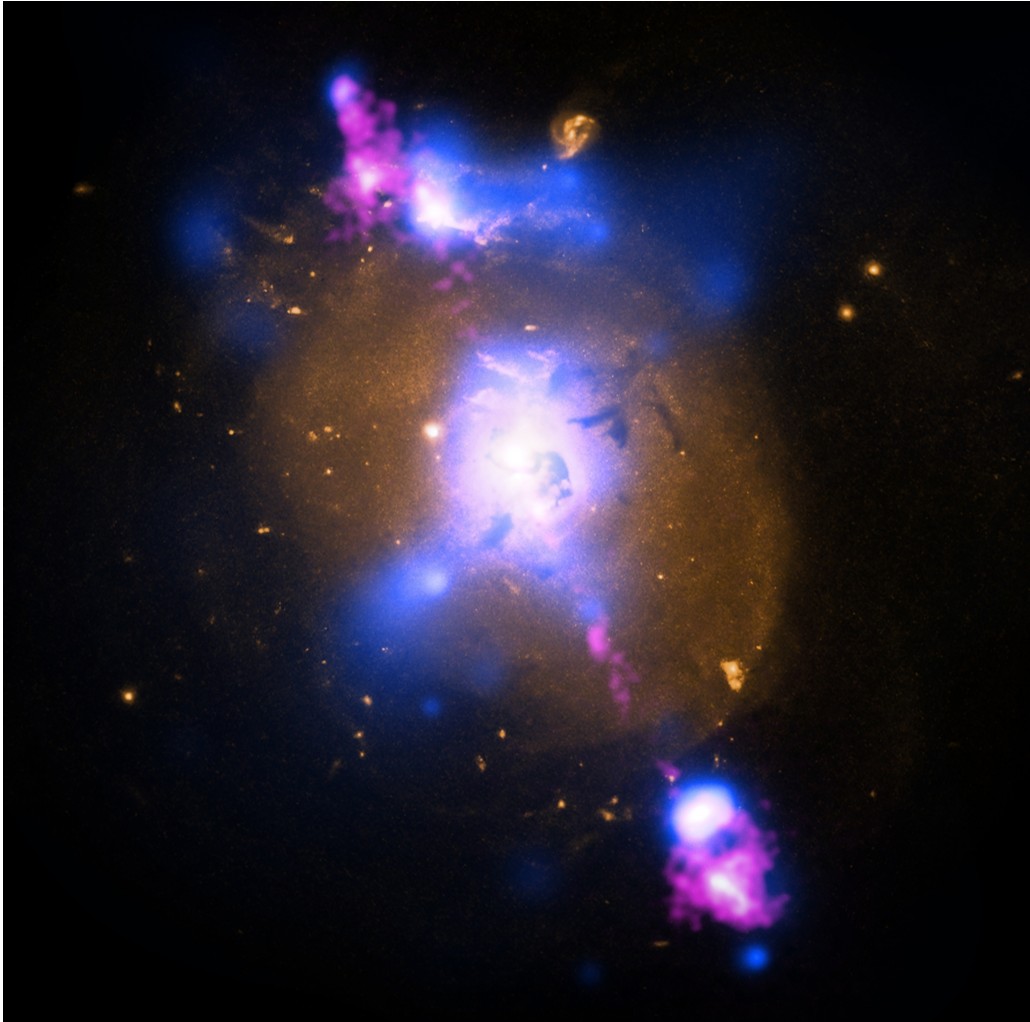

**Figure 1.** Composite image of radio-loud AGN 4C +29.30, with *Chandra* X-ray, *HST* optical and VLA radio observations [47,48]. Three different time periods of radio activity have been detected studying the radio-jet emission in this galaxy, which can be related with the galaxy interaction history (with clear signatures observed in the optical) and the feeding of the central SMBH. Credit: X-ray: NASA/CXC/SAO/A.Siemiginowska et al; Optical: NASA/STScI; Radio: NSF/NRAO/VLA

NGC 3801 is one of the few radio-loud AGN where shock-induced X-ray emission due to jet-gas interaction was observed, and it seems to be in a rare evolutionay stage where the enhanced star formation from a merger event has declined and a young powerful AGN activity is taking place [49,50]. In this scenario, early stages of the radio source expansion such as in Fanaroff-Riley I objects [51] have a more dramatic effect on the interstellar medium (ISM) when compared to larger scale extended sources such as FR II objects. The decline of star formation and the role of radio feedback into heating the gas seem to be important mechanisms to drive this type of objects in the 'red sequence', or 'red-and-dead' elliptical type galaxies.

The relation between radio-loud AGN and galaxy interactions seems to hold true also for higher redshift sources. Making use of high sensitivity *HST* imaging, Chiaberge et al. [52] studied the dependence of merger fractions in radio-loud and radio-quiet AGNs for redshifts $1 < z < 2.5$, when the AGN activity has peaked over cosmic history. The authors find that $92\%^{+8\%}_{-14\%}$ of the powerful radio-loud galaxies ($P_{1.4\,\mathrm{GHz}} > 10^{30}\,\mathrm{erg\,s^{-1}\,Hz^{-1}}$) are merging systems, while only $38\%^{+16\%}_{-15\%}$ of the radio-quiet galaxies present merger signatures.

Another possible influence of mergers into the radio loudness of active galaxies is its impact in the spin of the SMBH. As discussed in Chiaberge et al. [52], if gas accretion

resulting from a merging galaxy occurs with a similar angular momentum axis as that of the SMBH, it may result in an increase in the SMBH spin velocity, while if the accretion happens in a different direction, the SMBH spin can be slowed. These different orientations may then affect the nature of the radio-loudness of the AGN, as it is expected that faster spinning SMBHs generate more powerful radio jets, and the radio-loud/quiet characteristic could be explained by different accretion and SMBH spins [53,54]. But, of course, this effect is expected to be more prominent at high redshifts ($z \gtrsim 2$), as gas-rich mergers where more common in the younger Universe [55].

The close environment of radio-loud AGN, within its dark matter haloes (of radius $\sim 1\,\mathrm{Mpc}\,h^{-1}$) is denser than that of radio-quiet galaxies overall, but this also happens when the galaxies are matched by stellar mass and redshift [56,57]. The fact that galaxies with high stellar masses do not necessarily host a radio-loud AGN indicates that the presence of radio-loud jets could be influenced by the environment of the host galaxy [58]. In response, AGN radio jets seem to be important to control the star formation within the dark matter halo of its host galaxy, resulting in slower stellar mass build-up process, and the hosts would have lower stellar masses if the radio-mode feedback would not be present [59].

## 3. Radio-Mode Feedback in Galaxies

As pointed out in the the Introduction, radio jets from giant elliptical galaxies at the center of clusters can extend up to Mpc scales and produce strong feedback in the surrounding medium (e.g., [2,5]). But, as this topic is being discussed elsewhere in this volume, we focus here on the effect of radio-mode feedback inside galaxies. In this section we present some of the characteristic features of the radio-mode feedback and how the interactions between the relativistic jets and the gas in its path manifest itself within active galaxies.

### 3.1. Complex Gas Kinematics

The relativistic jets present in radio-loud AGN, when coupled with the circumnuclear gas either in the narrow-line region (within the radius of influence of the AGN ionizing radiation) or further out in the ISM (within the inner few kpcs), can be responsible for the heating and acceleration of this gas resulting in outflows. These outflows are usually detected in the turbulent ionized emitting gas by measuring its kinematics and isolating the kinematic components linked to the AGN feedback (e.g., [60–62]). The superposition of one or more kinematic components to that originating in gas rotating in the galaxy potential can result in a complex ionized gas spectrum, with many components, making the decomposition process hard to constrain [63]. With the advent of better spatial and spectral resolutions IFUs, the number of resolvable kinematic components has increased, and the interpretation of these components and their origin has also increased in complexity.

The case of the Seyfert 2 galaxy NGC 7130 is a clear example of such complexity, as discussed in Comerón et al. [17], using MUSE narrow-field adaptive optics observations. Even though the radio jet in this Seyfert galaxy is not powerful, one or maybe two outflowing ionized gas components seem to be interacting with the radio jet, as revealed by the very detailed emission-line decomposition performed by the authors, which comprises a total of nine components, with six being connected to outflows. It is important to note that the fitting of several components should be statistically justified so that the decomposition does not introduce artificial components into the emission-line fit.

Usually a rotation component is observed in the gas velocity field of radio-loud AGN, but alongside more complex kinematics, as illustrated in the study performed in the MURALES survey [64,65], in which a sample of 37 radio galaxies from the Third Cambridge Catalog (3C) were observed with the MUSE IFS. As shown in Figure 2, the ionized gas velocity maps of these radio galaxies appear to show some rotation, but present a much more disturbed pattern than the expected "web diagram" characteristic of the isovelocity curves of ordered rotation. Although these complex kinematics may not be the case for all radio-loud AGN, they are commonly observed, as several other studies of individual

galaxies have also shown, such as in 4C +29.30 [16], Cygnus A [66] and UGC 05771 [67], and are usually caused by the interaction between the radio jet and the circumnuclear gas.

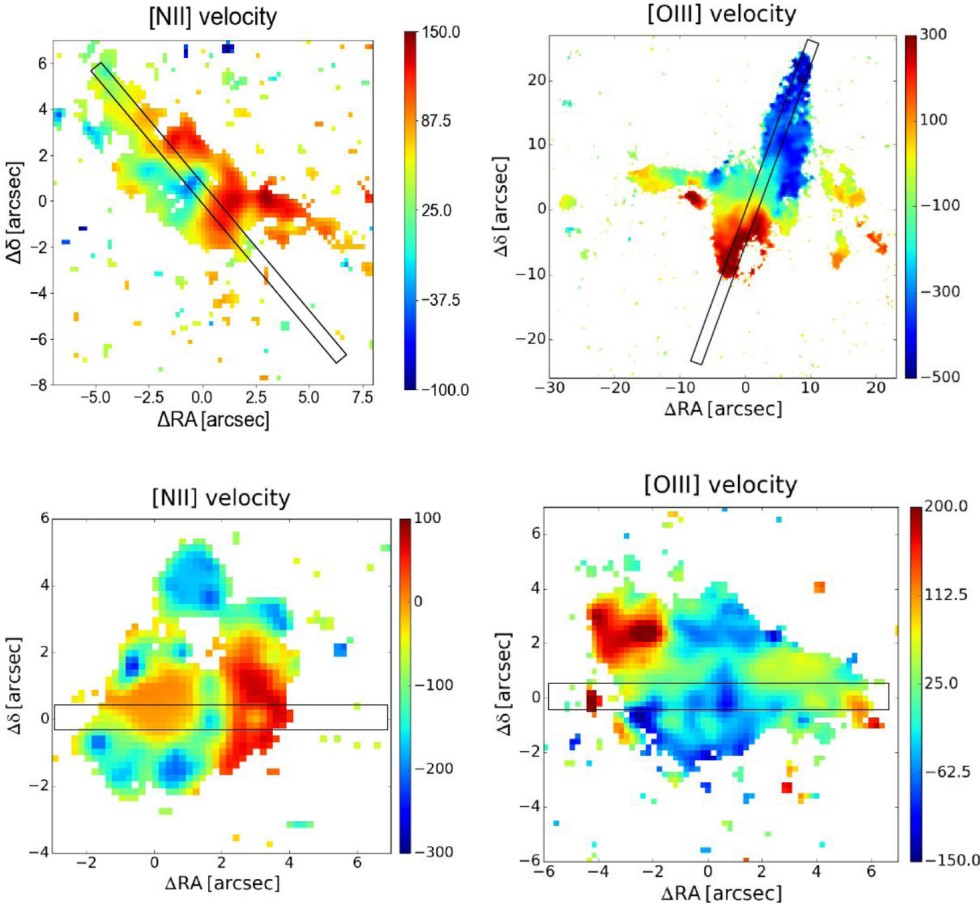

**Figure 2.** Velocity maps in the indicated emission-lines of the radio-loud AGNs 3C 076, 3C 079, 3C 018 and 3C 017 (clockwise from top left, [64,65]). Although some rotation seems to be present, disturbed kinematics dominate the velocity field of these galaxies, attributed to the interaction of the ambient gas with the radio jet. Velocity units are km s$^{-1}$.

An alternative method to track different kinematic components is to probe the velocity field along channel maps, which is possible when using IFS observations. Arp 102B is one case for which the channel maps can aid in the interpretation of how the radio jet interacts with the surrounding emitting gas. In Couto et al. [68], we have used Gemini-GMOS IFU observations to analyze the ionized gas kinematics in this galaxy. H$\alpha$ channel maps, shown in Figure 3, indicate that a spiral arm-like structure correlates spatially with the radio jet, and the emitting gas is observed both in blueshifted and redshifted velocities. We have interpreted that these velocities trace the outflowing gas pushed by the radio jet oriented very close to the plane of the sky. The channel maps shows emission from the "walls" surrounding the outflow being pushed aside, seen in both blueshifted and redshifted velocities. The use of channel maps to interpret the relation between the jet and the gas in individual galaxies studies has been proven useful in the past years, as displayed in other papers by our group such as in Lena et al. [61], Schnorr-Müller et al. [69], Riffel et al. [70].

### 3.2. Radio Bubbles

When the jet collimation is lost due to interaction with dense gas or jet precession, sometimes a gas bubble is formed, and the inflation of the bubble by the outflow also impacts the ISM. A nuclear starburst leading to numerous supernovae explosions can also produce outflowing bubbles from galaxy centers. The bubble serves as a shock front

between the outflow and the ISM gas, and when it erupts several galaxy properties can be altered, including its chemical composition. Perhaps the most famous case is of our own galaxy. Bipolar bubbles at the Galaxy center have been observed in several wavelengths, including in the radio at 1.3 GHz with MeerKAT [71], and it is still debatable whether these bubbles originate from AGN or star formation feedback. A similar case where there is ambiguity in the feedback origin is NGC 3079 [72]. While studying the AGN-starburst composite galaxy NGC 6764, Hota and Saikia [73] also found an inconclusive origin for the feedback, and compared it to 10 other sources. These are all associated with AGN activity, but the radio and optical emission could be also affected by central starbursts. Pure starburst sources do not seem to present the same structures, indicating that AGN must be present to produce the large-scale bubbles.

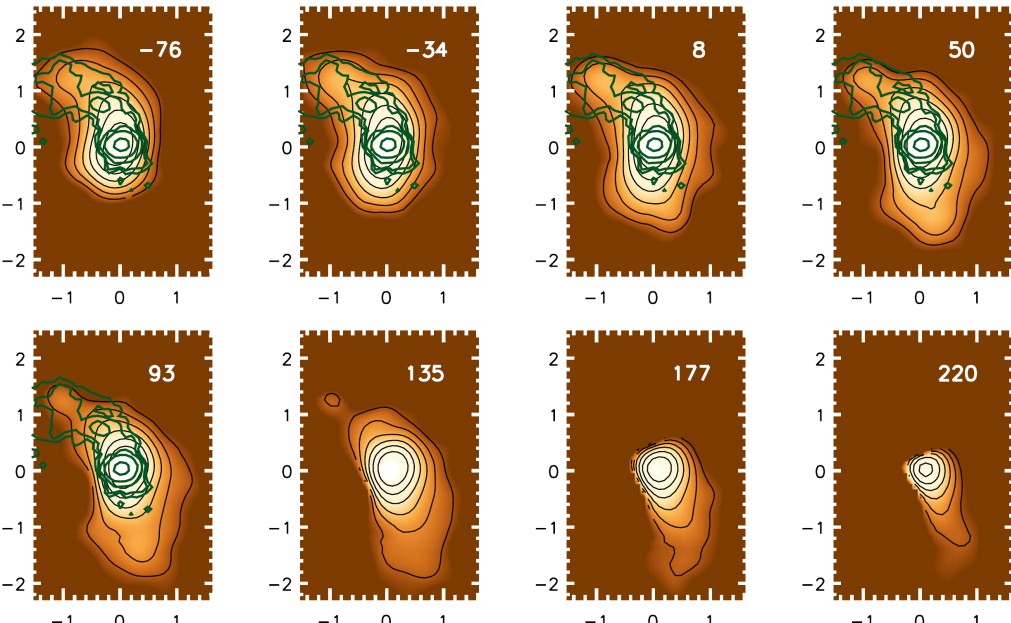

**Figure 3.** Channel maps along the Hα emission-line profile of the radio galaxy Arp 102B, with velocities displayed in white at the top right corner of each map (in units of $km\,s^{-1}$, [68]). The flux distribution maps show a spatial correlation with the radio jet, represented by the green contours from VLA observations at 8.4 GHz. We have interpreted this emission to be related to outflowing gas pushed aside by the jet moving close to the sky plane, thus displaying blueshifted and redshifted velocities originated in the gas surrounding the radio jet. X and Y-axis are in arcsec and are centered at the galaxy nucleus.

*3.3. Molecular Gas*

Besides the effect on the ionized gas phase, that trace the hot and turbulent outflows, molecular gas can also show signatures of jet-gas interaction. As the ionized gas represents only a fraction of the total gas mass, which is dominated by the molecular gas in the galaxy inner regions, the outflow gas mass should also be dominated by the molecular gas, specially in low AGN bolometric luminosities [74]. Mass outflow rates observed in molecular gas are about 2–3 orders of magnitude higher than those traced in the ionized gas phase ($\sim 1000\,M_\odot\,yr^{-1}$ as compared to $\sim$1–10 $M_\odot\,yr^{-1}$, respectively) in AGN with bolometric luminosity $\sim 10^{46}$ erg s$^{-1}$ (e.g., [75]). As a comparison, star-formation rates of 10–100 $M_\odot\,yr^{-1}$ (see Figure 3 in Fiore et al. [75]) are estimated for AGN in the same luminosity range, displaying that these outflows must indeed originate from AGN activity.

As shown in a study of the galaxy IC5063 by Dasyra et al. [76], by estimating the internal and external pressures of the molecular clouds one can infer the impact of the radio jet in the star formation processes within the galaxy, where both suppression and enhancement can simultaneously happen. In NGC 613, disturbed gas can be traced in the

nucleus due to molecular outflows mainly boosted by the radio jet [77]. While modeling the molecular gas outflows in the young radio galaxy 4C 31.04, Zovaro et al. [78] could reproduce the observed kinematics by assuming that the gas is being pushed and expanded by the radio jet in an energy bubble while generating shocks within this bubble, originating the observed $H_2$ emission. As observed (and discussed above) for the ionized gas phase, rotation is also usually observed in the molecular gas phase, but with distortions commonly found in the inner few hundred pc, as analyzed by Ruffa et al. [79] in a sample of six low excitation radio galaxies using ALMA observations. However, in the case of this sample, the authors interpreted that these non-rotating components are related to inflowing gas, since they seem to be correlated with structures such as spirals or bars, known for being mechanisms causing the gas to lose angular momentum to reach the nucleus and feed the SMBH. This illustrates that other signatures besides the gas velocities, should be used in the search of outflows, such as velocity dispersion and line ratios and their relation to the observed kinematics. Other cases of galaxies presenting jet-gas interaction signatures through the analysis of their molecular gas kinematics include NGC 1377 [80], ESO 420-G13 [81] and NGC 7319 [82], among others.

### 3.4. Models and Simulations

Detailed 3D hydro-dynamical simulations indicate that indeed the gas can be perturbed by the radio jet within the host galaxy. Not only is the jet responsible for disturbing and shaping the emitting gas distribution, as discussed in Wagner and Bicknell [83], the path taken by the jet and its morphology is also affected by the inhomogeneity of the gas density. In this scenario, the jet collimation, its power, and how it spatially couples with the gas are important parameters to estimate the feedback energetics. One example of such simulations is displayed in Figure 4, where the gas density distribution is shown while the radio jet evolves with time: the jet carves its way through the gas, pushing it both aside and forward to larger distances while heating it and possibly creating shock ionization. How easy and straight is the path of the jet through the gas depends on its density and porosity distributions. We refer the reader interested in learning more about simulations of jet-gas interactions to papers by Mukherjee et al. [84,85], Talbot et al. [86], Meenakshi et al. [87] for further details.

### 3.5. Signatures of Shocks Due to Radio Jets

During the interaction between the relativistic jet and the surrounding gas, gas ionization through fast shocks can occur, with the excited gas emitting characteristic spectra that provide information about the physical parameters of the shock ionization. Observed line ratios in the narrow-line region (NLR) indicating the presence of shocks usually lie in the Low-Ionization Emission Line Region (LINER, [88]) part of optical diagnostic diagrams, such as the well known BPT diagrams [89], depending on parameters such as the shock velocity and gas density [90,91]. Although photoionization models have a considerable overlap with shock models in the BPT diagrams, high values of low-ionization line ratios such as [N II]/H$\alpha$ and [S II]/H$\alpha$ are usually interpreted as tracers of shocks when gas emission is observed due to jet influence.

The presence of shocked gas also commonly correlates with broader emission lines, which trace the increase of gas turbulence. This is the case of 3C 293 [62], where jet-driven outflows are detected along the radio emission and two kinematic components are needed to reproduce the ionized gas emission-line profiles. A clear increase in the [N II]/H$\alpha$ and [S II]/H$\alpha$ line ratios is observed in the regions where a broader component is required to reproduce the emission lines (see Figure 6 in [62]), indicating that the shock velocity increases in the regions where the highest ratios are observed.

Another interesting case is the gas excitation observed in 3C 33 [92]. As displayed in Figure 5, velocity dispersion increases in a region surrounding the nucleus, perpendicular to the radio jet (a discussion that we will explore further in Section 5), in the same orientation where high velocity residuals are observed when a rotating disk model is subtracted

from the Hα velocity field. This correlation indicates that this region is dominated by a non-rotating kinematic component. Using channel maps of the [N II]/Hα line ratio, we could detect the increase of this line ratio with the velocity residuals, since the regions presenting the highest blueshifted and redshifted residuals show high [N II]/Hα values with the same velocity shift. Other shock-driven outflow signatures, such as an increase of gas temperature (obtained using the [O III]$\lambda 4959 + 5007/4363$ line ratio) and density, strengthen the scenario that the excitation of this component is (at least partially) due to kinetic-mode feedback.

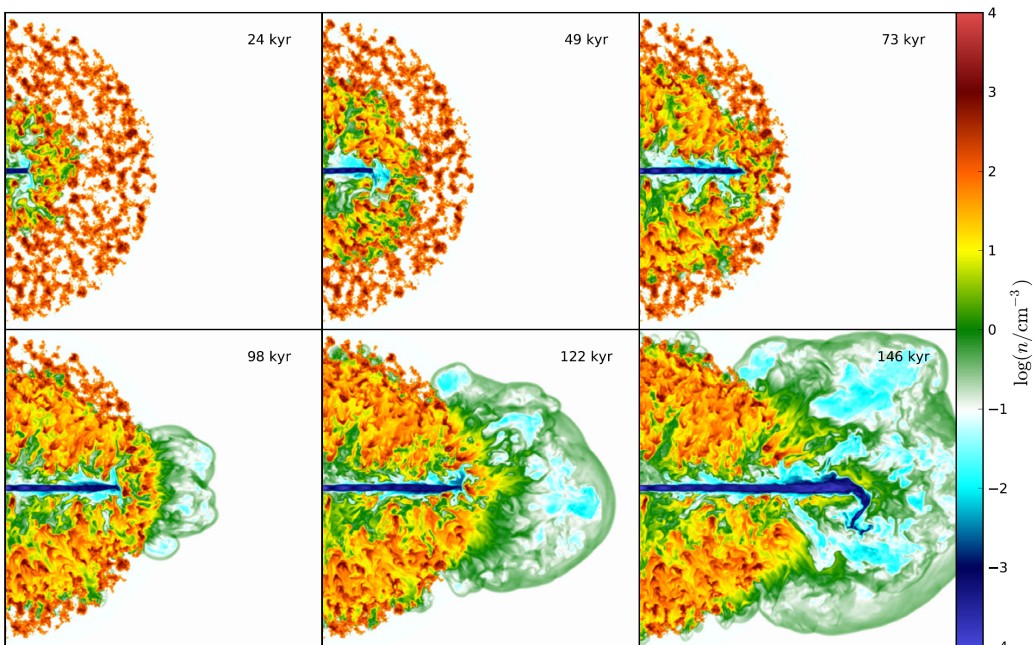

**Figure 4.** Figure 2 from Wagner and Bicknell [83], showing the time evolution of the jet-gas interaction in one of the simulations performed in the paper. Colors represent the different gas densities as the jet makes its way through it, being deflected towards regions of less resistance and finally creating a bubble of expanding gas. Some of the dispersed gas may reach velocities of up to $\sim 1000\,\mathrm{km\,s^{-1}}$.

Shock excitation has been found in several other objects presenting radio jets, such as M51 [60], Coma A [93], the Beetle galaxy [94], PKS B1934-63 [95], 3C 320 [96], 4C 31.04 [78], 3C 433 [97], J1220+3020 [98], Cygnus A [66], among others. Shock excitation has been found not only in the ionized gas phase, but also sometimes in molecular or neutral gas. High gas velocities of up to $\sim 1000\,\mathrm{km\,s^{-1}}$, or even higher, are commonly found in these sources, and an increase of velocity dispersion is also a good tracer of the regions presenting shocks.

*3.6. Feedback Power and Scaling Relations*

After characterizing gas outflows due to the AGN jet feedback, it is useful to estimate its kinetic power in order to quantify its impact on the host galaxy and compare it with models and other sources of feedback. The kinetic power of the outflow attributed to the kinematic disturbance produced by the radio jet can be calculated via [99]:

$$\dot{E} = 6.34 \times 10^{35} \frac{\dot{M}_{out}}{2} \left(v_{out}^2 + 3\sigma^2\right),\tag{1}$$

where $\dot{M}_{out}$ is the mass outflow rate, $v_{out}$ is the deprojected outflow velocity and $\sigma$ is the outflow velocity dispersion. The mass outflow rate is dependent on the assumed outflow geometry, and can be expressed through:

$$\dot{M} = 1.4\, n_e\, m_p\, v_{out}\, A\, f\,,\tag{2}$$

where $n_e$ is the gas density, $A$ is the assumed geometric cross section area of the outflow, $f$ is the filling factor within the outflow volume and $m_p$ is the proton mass ($m_p = 1.7 \times 10^{-24}$ g), while the factor 1.4 accounts for the contribution of elements heavier than hydrogen.

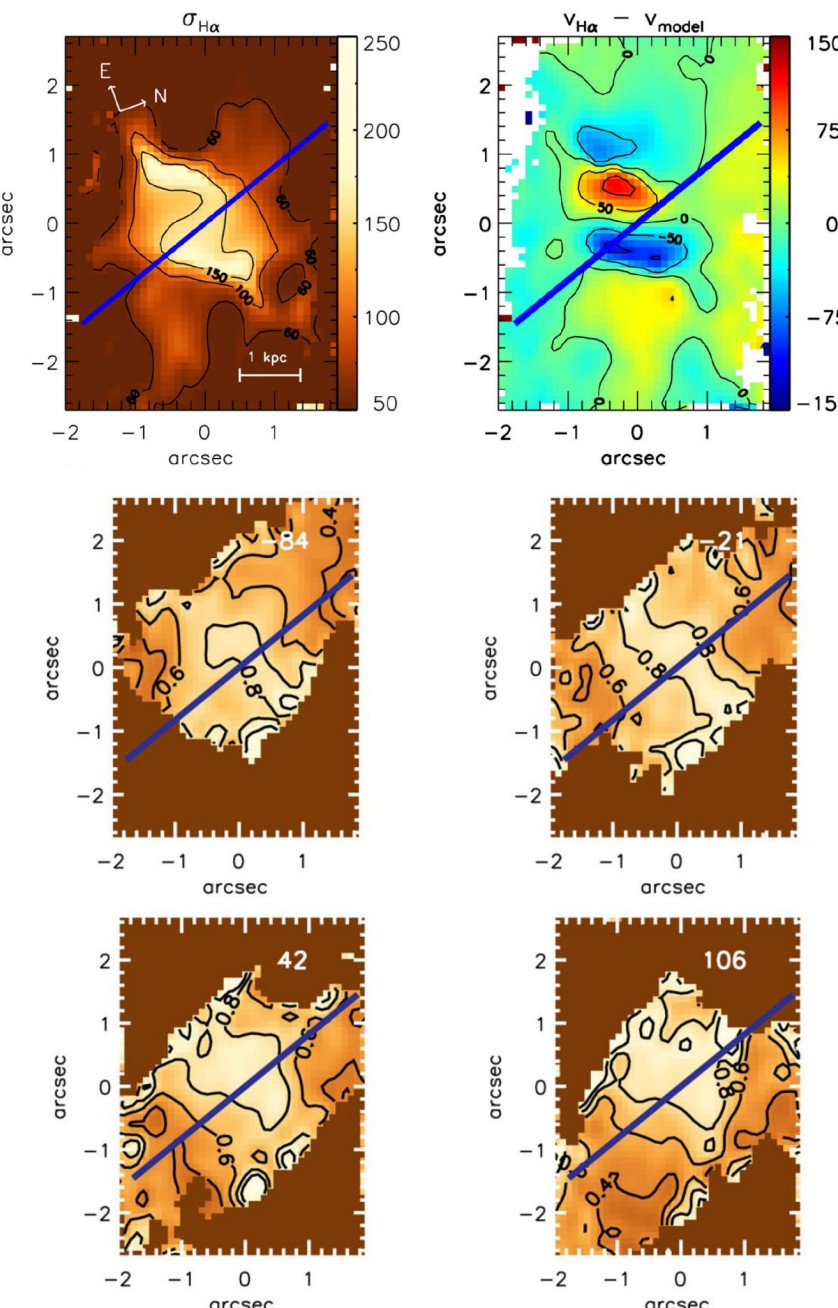

**Figure 5.** Gemini-GMOS IFS observations of the radio galaxy 3C 33. H$\alpha$ velocity dispersion (top left), velocity residuals (top right, observed velocity field minus disk-like rotation model) and [N II]/H$\alpha$ line ratio channel maps (middle and bottom panels), adapted from Couto et al. [92]. Channel map velocities ($-84$, $-21$, 42 and 106 km s$^{-1}$, clockwise from top left) are displayed in the top right corner of the panels. The blue line displays the large-scale radio jet orientation. An increase in the velocity dispersion is observed in the same region where velocity residuals are highest. The same velocities are spatially correlated with an increase of the [N II]/H$\alpha$ line ratio (delineated by contours), indicating that shock-driven outflows are present in these regions, probably due to the jet-gas interaction.

The calculations above are not restricted to radio-loud AGN, but are general to any type of outflow. For the ionized gas, the gas density is usually estimated using the

[S II]$\lambda 6717, 31$Å line ratio, and the filling factor can be obtained using a measured hydrogen emission-line luminosity, such as H$\alpha$ [100]:

$$f = 2.6 \times 10^{59} \frac{L_{41}(\mathrm{H}\alpha)}{V\, n_3^2}\,, \tag{3}$$

where $L_{41}(\mathrm{H}\alpha)$ is the H$\alpha$ luminosity in units of $10^{41}$ erg s$^{-1}$, $V$ is the assumed geometry volume, and $n_3$ is the gas density in units of $10^3$ cm$^{-3}$.

Scaling relations using the estimated mass outflow rates $\dot{M}$ and outflow kinetic power $\dot{E}$ of AGN-driven feedback have been the object of studies in the past few years. Fiore et al. [75] investigated the relation between the bolometric luminosity and both the mass outflow rate and outflow kinetic power for 94 AGN covering a large range of redshifts (from local Universe up to $z \sim 6$) compiling observational data from the literature and homogeneously calculating the mass outflow rates and powers. Ionized and molecular gas show clear correlations between $L_{bol}$ and both $\dot{M}$ and $\dot{E}$, with somewhat higher $\dot{M}$ and $\dot{E}$ for the molecular gas, that seems to reach the coupling efficiency in the range 1–10%$L_{bol}$, as required in models (e.g., [101,102]) to have a significant impact on the host galaxy (e.g., by pushing the gas out of the galaxy and halting star-formation) while for the ionized gas only $\approx$30% of the galaxies reach this efficiency.

More recent studies (e.g., and references therein [103,104]) on kinematic feedback via ionized gas outflows have obtained higher gas densities and resulting lower powers than those estimated in Fiore et al. [75] for most AGN. In some cases, like in the radio-galaxy 4C +29.30 [16], the outflow power can reach a few percent of the AGN power $L_{AGN}$, implying strong feedback, but, in most cases, this power is below 1%$L_{AGN}$. The kinematic feedback is usually present, heating and disturbing the gas – a "maintenance mode feedback", but not high enough to push the gas out of the galaxy or immediately halt star formation. On the other hand, AGN feedback occurs not only via outflows, with recent model estimates suggesting that at most 20% of the AGN feedback is in kinetic form (e.g., [105]).

The relations between $\dot{E}$ and $L_{AGN}$ and $\dot{M}$ and $L_{AGN}$ discussed above seem to apply both to radio-loud and radio-quiet AGN. In the case of the latter, Villar Martin et al. [106] has shown that, for a sample of 13 nearby ($z < 0.2$) radio quiet QSOs, most of which showing signatures of interactions, 10 had extended radio emission. In addition, they found this radio emission was correlated with the optical H$\alpha$ emission, indicating jet-gas interaction. Thus radio-mode feedback is also present in radio-quiet AGN, when the jet is spatially coupled with the ISM gas.

## 4. Statistical Studies and Higher Redshift Sources

There are a number of statistical studies of AGN feedback effect in general (not restricted to radio AGN) on host galaxies properties. Wylezalek and Zakamska [107] gathered a large sample of radio-quiet AGN at $z < 1$ with SDSS observations to investigate the relation between AGN feedback and star formation quenching. Measuring the [O III]$\lambda 5007$Å velocity width as an outflow tracer, the authors find no correlation between mass outflow rates and star formation rates; however, in galaxies with high specific star formation rates (sSFR), they found a negative correlation between outflow strength and sSFR, indicating that in these galaxies, with the highest gas content, there is quenching due to AGN outflows.

Studying a much larger sample of SDSS AGN, Mullaney et al. [108] performed multiple component fit to the optical emission lines and found that AGN with moderate radio luminosities ($L_{1.4\,\mathrm{GHz}} = 10^{123} - 10^{25}$ W Hz$^{-1}$) present the most disturbed gas kinematics with the highest incidence of extremely broad [O III]$\lambda 5007$Å emission. This suggests that young and compact radio sources seem to more effectively drive gas turbulence in the host galaxies then powerful extended jets of the most radio luminous sources.

At higher redshifts, we cite the work of Delvecchio et al. [109], where they performed multi-wavelength analysis of 7700 radio selected AGN from the COSMOS field with VLA observations with redshifts up to $z \lesssim 6$. They divided the sample in moderate-to-high and

low-to-moderate radiative luminosity AGN (MLAGN and HLAGN, respectively). The authors found that the HLAGN have systematically higher radiative luminosities and the AGN power occurs predominantly in radiative form, while for MLAGN the AGN power has also a large mechanical component. They also find that at $z < 1.5$, MLAGN reside in more massive and less star-forming galaxies compared to HLAGN. At $z > 1.5$, the opposite is found: the HLAGN seem to occur in more massive galaxies. The authors interpret their findings as evidence of downsizing, with the most massive galaxies triggering AGN earlier than the less massive galaxies. At lower redshifts, the HLAGN fade to MLAGN.

## 5. Feedback Perpendicular to the Radio Jet Orientation

Even though gas outflows are usually expected to be found along the AGN radio jet, this is not always the case. Recent studies have been showing different orientations between the radio jet axis and the direction of an observed increase of the gas velocity dispersion related to outflowing gas. In many cases the direction of the enhanced velocity dispersion is approximately perpendicular to that of the radio jet. Perhaps the first observational evidence of such orientation in an IFS analysis was the case of Arp 102B [68]. In this radio galaxy, we observed an increase in the gas velocity dispersion in the [O III]$\lambda$5007Å emission-line, close to the nucleus, approximately perpendicular to the radio jet. The jet then bends with increasing distance to the nucleus.

Ionized gas velocity dispersion maps of Arp 102B and other galaxies presenting enhanced values approximately perpendicular to radio jets are shown in Figure 6. Arp 102B, NGC 1386 [61], NGC 5643 [19], 3C 33 [92] and NGC 3393 [110] show very clear signatures of the misalignment between the radio jet and the extended high velocity dispersion emission, very close to perpendicularity. In a very detailed study of this phenomenon, Venturi et al. [19] used VLT MUSE observations to trace the outflowing gas in four galaxies displaying mismatch relative to the orientation of the radio jets, including NGC 5643, shown in Figure 6. Extended emission ($\gtrsim 1\,\mathrm{kpc}$) is observed with increased velocity dispersion (W70 $\gtrsim$ 800–1000 km s$^{-1}$) perpendicularly to the jet. The authors also found that the gas excitation in these regions is consistent with shock ionization, similarly to the results we have found in 3C 33 (as discussed in Section 3.5, where higher shock-related line ratio values ([S II]/H$\alpha$ in their case) are observed. Further evidences of this phenomenon was observed in the AGNIFS survey [18], comprising 30 local AGN observed with GMOS-IFU, in which outflows perpendicular to the ionization axis (via enhanced W80 values) were detected in 7 sources out of 21 that present outflows. The same was found in the MURALES survey [64], where at least 6 of the initial 20 observed galaxies present some indication of turbulent ionized gas perpendicular to the radio jet. NGC 5929 [70] is another case where the velocity dispersion of the ionized gas is found to be enhanced perpendicularly to the radio jet (see their Figure 8).

As discussed in Venturi et al. [19], the galaxies presenting outflows perpendicular to the radio jet are usually hosts of either compact or low-power radio jets (with some exceptions, like 3C 33), and thus the bulk of the jet emission is close to the nucleus, within the inner kpc or less. These jets are also not perpendicular to the galaxy disk, with low inclination angles between these structures, resulting in a larger impact of the jet in the ISM. Galaxies that present jets oriented perpendicularly to the galaxy disk plane do not display the same behavior. The spatial correlation of the regions with high gas velocity dispersion with high line ratios related to shocks (as shown in Figure 5) indicates that these outflows are shock-driven due to the jet interaction with the galaxy disk gas. In galaxies with large-scale jets, usually FR II objects, with lobes extending tens of kpcs from the galaxy nucleus, the jet could have already cleared much of the gas in its path. Also, high-power jets ($\gtrsim 10^{45}$ erg s$^{-1}$) penetrate the clumpy ISM and dig their way out through the gas easier when compared to low-power jets [83,85].

A similar scenario has been reproduced by radio-mode feedback models. The jet power and its inclination relative to the galaxy disk determines how the jet affects the gas in the NLR and ISM, as discussed in Mukherjee et al. [85]. Jets inclined closer to the

disk influence the gas more than when its orientation is perpendicular to it, generating shock-driven wide-angle outflows along the disk minor axis and enhancing turbulent dispersion along the disk. As shown by the gas density evolution along the jet axis in Figure 7, with a jet axis inclination of 45° with the disk plane, the jet interacts strongly with the gas, and the denser gas (when compared to higher disk latitudes) deflect the jets, which is decelerated. The direction of least resistance is where the turbulence is propagated, which is along the direction perpendicular to the radio jet.

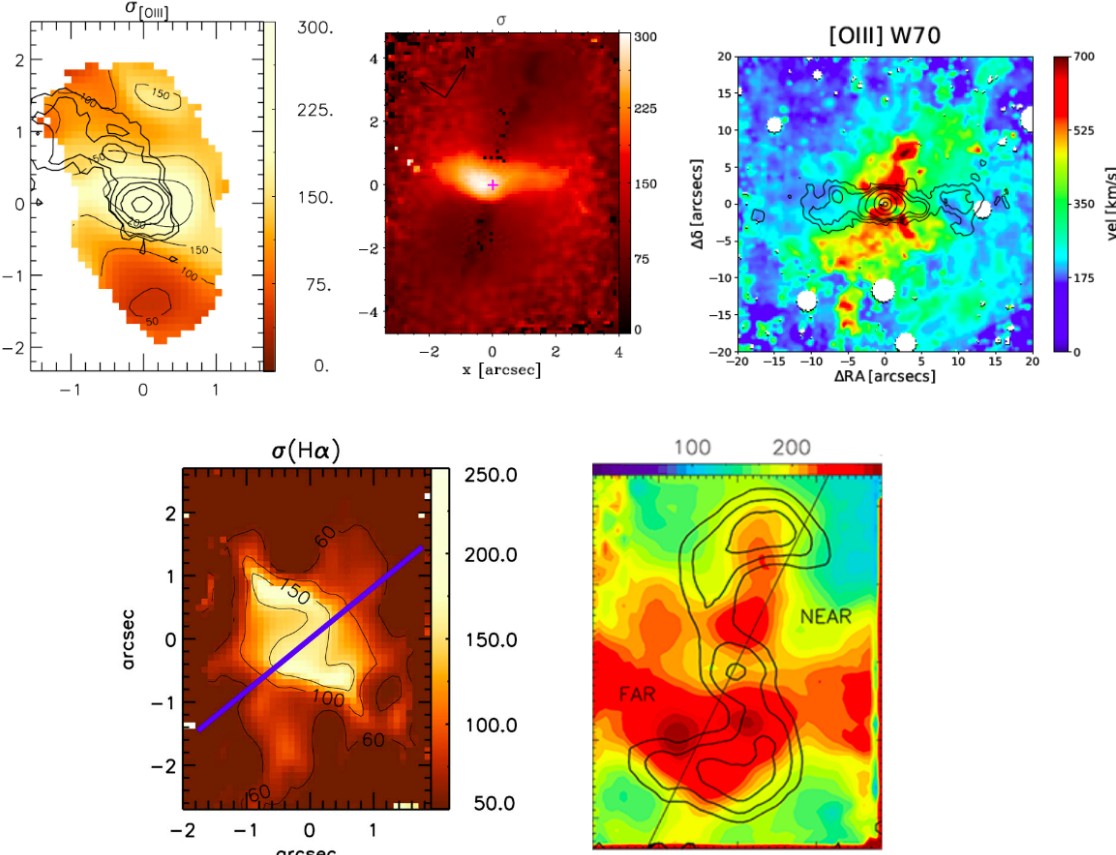

**Figure 6.** Gas kinematics of several active galaxies displaying increased gas velocity dispersion–attributed to outflowing components–perpendicularly (or close to being perpendicular) to the radio jet orientation. Top left: Arp 102B Gemini GMOS-IFU [O III] velocity dispersion map, with VLA 8.4 GHz contours displaying the radio jet emission [68]. Top middle: NGC 1386 GMOS-IFU [N II] velocity dispersion map, with the dark region along north-south axis tracing the jet emission (not shown here) and enhanced integrated flux [61]. Top right: NGC 5643 VLT-MUSE [O III] W70 map showing highest values perpendicular to the radio jet shown via VLA 8.4 GHz emission contours [19]. Bottom left: 3C 33 Hα velocity dispersion map, with the large scale VLA 1.4 GHz radio jet orientation represented by the blue line [92]. Bottom right: NGC 3393 GMOS-IFU [N II] velocity dispersion map also displaying VLA 8.4 GHz radio emission as black contours [110]. Velocity dispersion and W70 units are in $km\,s^{-1}$.

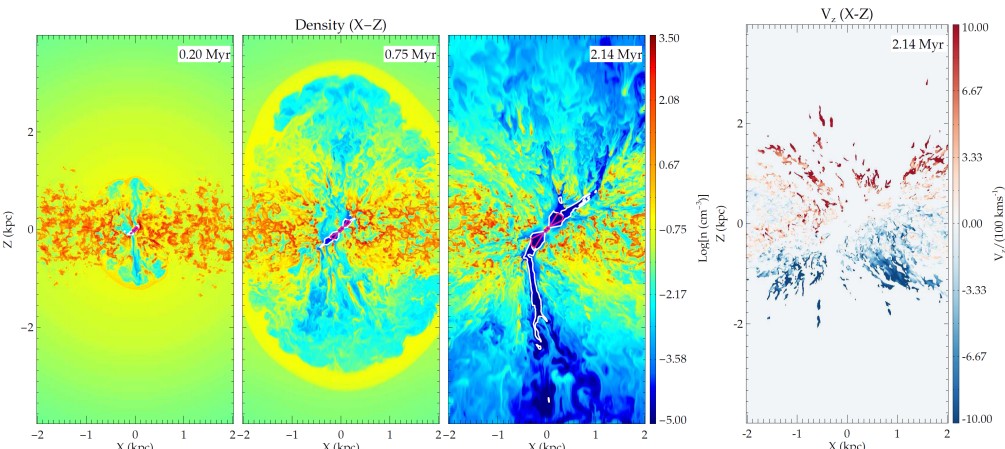

**Figure 7.** One of the simulated interactions between the jet and the gas by Mukherjee et al. [85], in this case with in 45° inclination between the jet axis and the galaxy disk plane. Left panels display the time evolution of the gas density along the radio axis plane. The right panel shows the projected velocity of the gas, where a clear enhanced component is observed perpendicularly to the jet. As a result of the enhanced interaction with the ISM, in comparison to scenario with the jets being launch perpendicularly to the disk, the jet is decelerated and generate higher turbulence dispersion, mainly towards the path of less resistance perpendicularly to it.

## 6. Maintenance Mode Feedback-Red Geysers

In Comerford et al. [111], the authors have identified in the MaNGA/SDSS galaxy survey (Mapping Nearby Galaxies with the Appache Point Observatory of the Sloan Digital Sky Survey) [112] a number of low-luminosity AGN in radio using the NVSS and FIRST radio surveys, many of which were not detected in the optical observations. They separated the sample in radio-mode and radio-quiet AGN and, comparing the two subsamples, found that the radio-mode ones are preferably hosted by early-type galaxies, have older stellar population and lower star-formation rates. Such galaxies are sometimes referred to as being "red and dead", and in Comerford et al. [111] the authors suggest that feedback from radio jets could be the source of the suppression of star formation.

But the authors also point out that one caveat of the conclusion above is the fact that early-type galaxies have less gas in the inner region, implying in low mass accretion rates to the nuclear SMBH, what leads to advection-dominated accretion flows (e.g., [113,114]). It is well known that such accretion flows favor the formation of radio jets. Thus, instead of radio jets being the source of feedback that would lead to red and dead galaxies, they are more a consequence of the low accretion rates to their SMBH, resulting from the scarcity of gas in the nuclear region of early-type galaxies. In addition, while radio activity in AGN are short-lived phenomena a few million years old, continuum optical and IR galaxy features, as well as the central stellar populations are hundreds million to billion years old, which means that the radio feedback has little to none effect in these structures. On the other hand, there is still mass-loss due to the process of stellar evolution that could still lead to some star formation that seems not to be there, and this could indeed be attributed to feedback—even if mild, produced by low-power radio jets, precluding new star formation. It is also important to take into consideration recurrent radio activity where relic emission is observed, mainly in merger remnants such as Centaurus A or NGC 3801. In these galaxies, the older activity could indeed have had an impact in the past star formation history of the host galaxy.

Low-power jets can produce a mild outflow, producing what is being called "maintenance mode feedback", observed also in a class of galaxies that have been called Red Geysers, in which a low-luminosity AGN seems to be the source of large-scale but mild outflows that were discovered in observations with the MaNGA [20]. The Red Geysers frequently show a radio-source at the nucleus [21], and recent optical IFS of the inner

region at a resolution at the galaxies of a few 100 pc using the Gemini telescopes [115] have revealed nuclear outflows that are misaligned relative to the larger scale outflows discovered in the MaNGA survey. This misalignment has been argued by Riffel et al. [116] as being due to precession of a nuclear outflow/jet, as illustrated in Figure 8.

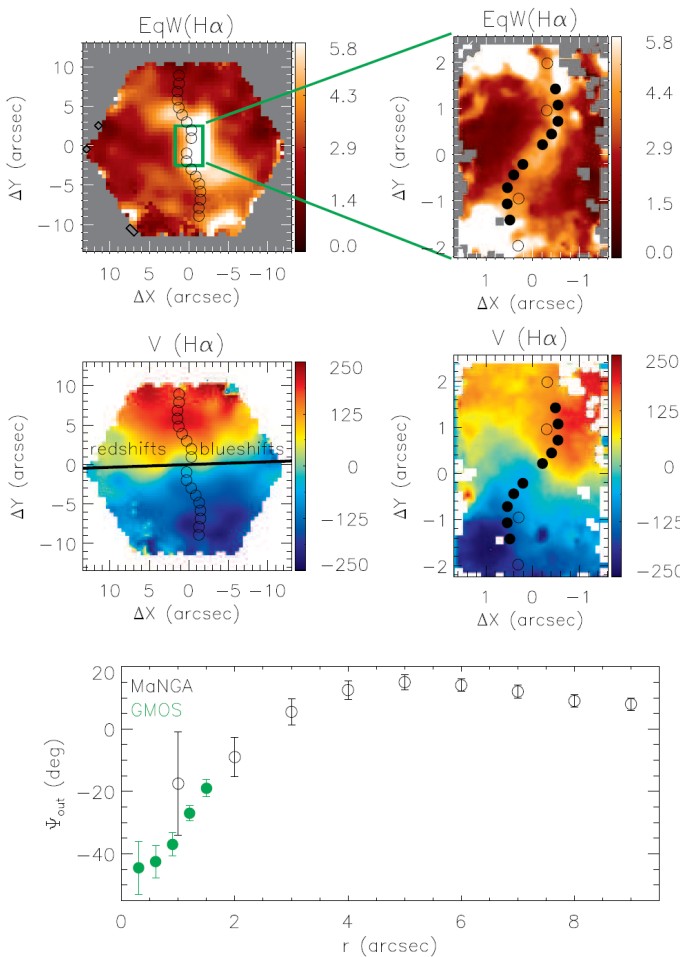

**Figure 8.** Outflow orientation of the prototype Red Geyser Akira (Figure 4 in [116]). SDSS MaNGA (left panels) and Gemini-GMOS IFS (right panels) display different scales of the galaxy (Gemini FoV represented by the red rectangle in the top left panel), with top and middle panels showing the H$\alpha$ equivalent width and velocity maps, respectively. Circles represent the orientation of outflowing gas, after interpreting that these regions cannot be reproduced by disk-like rotation, where open and closed circles are using MaNGA and GMOS data, respectively. Redshifted and blueshifted velocities obtained from the stellar kinematics are identified, along the line of the nodes measured in the stellar velocity field. The bottom panel displays the outflow orientation relation with distance to the nucleus. The variation in outflow orientation is interpreted as precessing winds due to the orientation of the accretion disk in relation to the SMBH spin.

Using MaNGA/SDSS datacubes of a sample AGN, Wylezalek et al. [63] has also shown that low-luminosity AGN can introduce mild kinematic disturbances over most of the galaxy extension as mapped via the $W_{80}$ kinematic parameter, that measures the width of line profiles comprising 80% of their fluxes. Using a control sample, Gatto et al. [in preparation] confirms the larger $W_{80}$ in AGN than controls down to the lowest AGN luminosities, configuring a mild, maintenance mode feedback.

Another possible scenario for the misalignment between the small and large scale ionized gas is the radio detection limit from observations using telescopes and arrays such as the VLA. Recent and future radio telescopes such as LOFAR and SKA will explore further the faint emission of older jet feedback, which may correlate with the large-scale

ionized gas orientation, while the gas closer to the nucleus is affected by younger and more compact activity. Example cases of these phenomena are NGC 5813 [117] and Mkr 6 [118]. A recent report of LOFAR observations by Webster et al. [119] describe a large sample of intermediate scale radio sources in low luminosity AGN at 150 MHz. As these new observations expand the samples of faint sources, we will be able to further constrain how the radio feedback interacts with its galaxy hosts.

## 7. Summary

AGN radio activity and its relation with the host galaxies has been the subject of detailed studies in the context of galaxy evolution. With the further development of IFS instruments, we have been able to trace and characterize gas outflows originated in jet-gas interactions in nearby radio galaxies. In this review we focused on the discussion of the interplay of radio-jets with the circumnuclear gas of radio AGN, displaying clear examples where these jets have an impact in the galaxies ISM. We showed how galaxy interactions seems to have an important role in the triggering radio activity, usually found in early-type galaxies, via the feeding of the nuclear SMBH.

Feedback in the form of outflows produced via the interaction of the radio-jet wih the circumnuclear gas is observed not only in ionized gas phases, but also in molecular gas, which should represent the bulk of the gas mass within the outflows, as we discussed in the context of AGN scaling relations. Jet-gas coupling and orientation of the outflows are important parameters to understand the produced feedback, as even low power radio jets can be well coupled with the gas and generate outflows. These outflows are observed both in the direction of the radio jet but also perpendicularly to it, usually traced by enhanced velocity dispersions perpendicular to the radio jet.

Although there are a few radio AGN that produce strong feedback on the host galaxies, most radio sources in the near Universe present outflow kinetic powers that do not reach 1% $L_{bol}$, and thus do not provide a strong and immediate impact on the host galaxy. Instead, they act to heat the ISM gas, preventing star formation, slowing the galaxy mass build-up process and limiting the stellar mass growth in a "maintenance mode" feedback.

The era of James Webb Space Telescope will allow us to perform similar spatially resolved studies at higher redshifts, revealing important information on AGN feedback across cosmic time. With the aid of incoming surveys such as 4MOST and WEAVE and the future IFS generation (such as ELT/HARMONI), we will be in better conditions to stablish the role of radio feedback on galaxy evolution, not only in high-power jets but also in low radio luminosity AGN, which also seem to be an important piece of the puzzle.

**Funding:** This research received no external funding.

**Data Availability Statement:** The data underlying this article will be shared on reasonable request to the authors.

**Acknowledgments:** The authors thank the anonymous referees for their valuable contributions on improving this review.

**Conflicts of Interest:** The authors declare no conflict of interest.

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
