# Peer review of "The Interplay between Radio AGN Activity and Their Host Galaxies"

_galaxies, doi:10.3390/galaxies11020047_

Round 1
Reviewer 1 Report
The authors have done a very good job in bringing out this review. Although the observation of jet-ISM interaction is old, since the VLA survey of Seyfert galaxies with jet-NLR correlations, it is indeed with the advent of IFUs that we are realising the importance of jets and its impact (or AGN Feedback) on the whole galaxy. Miles to go, still far from proving causality in any sample with a relic of past feedback and decline of star formation with matched-time-scales! I appreciate the highlight on the line-ratio study of the ionised gas detected orthogonal to the radio jet axis, opening up many possibilities to help make simulations more realistic. The paper is well-written and I only have some minor comments to improve it.
Section 1: Introduction:
For the sake of completeness, before focusing on molecular, ionised gas and radio jets, it would be good to refer to high ~1000 km/s velocity outflows seen in WSRT and signs of shock seen in X-ray imaging studies seen with the Chandra.
Section 2: Merger and AGN
In this section describing AGN luminosity and merger as a trigger, it may be a good idea to include the following two relations from ULIRG studies available through the ARAA review paper by Sanders & Mirabel (1996). There is an inverse relationship between internuclear distance and IR luminosity of ULIRGS. Furthermore, the AGN fraction increases with the increase of IR luminosity, demonstrating merger as a trigger and possible evolution of star formation to AGN activity.
Section 2.1 Individual Cases.
Perfect examples were given for the physical process we are interested in. I would like to draw author's attention to the case of NGC3801 where multi-wavelength data show a nice evolutionary sequence of merger, wind and jet, shock and outflows (Croston et al. 2007 and Hota et al 2012).
Section 3: radio mode feedback
In this section, I would suggest addition of the fact that radio mode feedback is not confined to narrow linear jet structures but also radio bubbles, specially so in sub-galactic scale. Some brilliant examples are Radio bubbles in our own Milky Way galaxy recently imaged with MeerKAT, NGC3079 (Cecil et al. 2001), Circinus Galaxy (Elmouttie et al. 1998) and a dozen or so examples compiled by Hota & Saikia 2006. Collimation of the jet is probably lost in many cases due to jet-ISM interactions and/or precession of the jet. Authors may add a couple of examples in a suitable place in this subsection.
Subsection 3.2:
Along with expressing that molecular gas dominates over the ionised gas fraction in the outflow, it may be useful to express them in numbers. Outflow rates being comparable to the star formation rate is a very convincing argument for AGN feedback.
Section 4: feedback perpendicular to the jet
This is not only a recent result but also seems to be the highlight of this review. Again, most studies in the past were either narrow band imaging or long-slit spectroscopy. Hence, IFU is going to lead the result in this new direction.
Instead of putting images from other sources that cover the full page of the paper, I would strongly advise for a schematic diagram highlighting this new kinematic component in a galaxy seen orthogonal to the jet.
Section 5: Red Geysers
The second paragraph describes an important caveat. Dividing samples with radio-loud and radio-quiet leads to a lot of confusing correlations or lack of correlations. Radio continuum feature, jet, is a short-term phenomena and most continuum optical/IR features are orders of magnitude older. Radio may be a few tens of million years old and an azimuthally symmetric optical feature in a galaxy may be a few 100 million years old. Radio jet and dust disks/rings seen in early-type galaxies do not represent feedback from jet reaching stellar population but must be from much earlier episodes of either jet/wind feedback, after the merger (e.g. Cen A, NGC3801).
In the next paragraph, on red-geysers, authors have mentioned precession of the jets for explaining misalignment of ionised gas on a larger and smaller scales and with the radio features. I think that these radio observations are limited to detection of compact/young/high-radio frequency features but when observed with low-frequency images like LOFAR/GMRT we may detect older episodes of jet feedback, coinciding with ionised gas detections at larger-scales. To explain some of the ionised gas seen orthogonal to jets, a speculation may be added that such things could also be due to episodic nature of jets which need not be in the same orientation as the recent jet but all episodes with in the ISM or stellar light distribution of the galaxy (NGC 5813; Randall et al. 2011, Mrk6: Mingo et al. 2011, Saikia & Jamrozy 2009). In fact LOFAR is now detecting large samples of galaxy 10-80 kpc scale low luminosity radio emission which are in between the kpc-scale radio emission in Seyferts and 300 kpc scale lobes of powerful radio galaxies (Webster et al. 2020). Future SKA may add to this study. Looking forward to the modified version. with best wishes...
Author Response
ANSWER TO REFEREE 1:
We thank the anonymous referee for the many useful suggestions, which helped us improve the manuscript. Changes to the manuscript are shown in red in the new version and answers to specific points are given below.
REVIEW 1
The authors have done a very good job in bringing out this review. Although the observation of jet-ISM interaction is old, since the VLA survey of Seyfert galaxies with jet-NLR correlations, it is indeed with the advent of IFUs that we are realising the importance of jets and its impact (or AGN Feedback) on the whole galaxy. Miles to go, still far from proving causality in any sample with a relic of past feedback and decline of star formation with matched-time-scales! I appreciate the highlight on the line-ratio study of the ionised gas detected orthogonal to the radio jet axis, opening up many possibilities to help make simulations more realistic. The paper is well-written and I only have some minor comments to improve it.
Section 1: Introduction:
Referee: For the sake of completeness, before focusing on molecular, ionised gas and radio jets, it would be good to refer to high ~1000 km/s velocity outflows seen in WSRT and signs of shock seen in X-ray imaging studies seen with the Chandra.
Authors: We do not enter into much detail regarding the gas outflows in the introduction, but we do agree that these works enrich the discussion. We have added a short text regarding this at the end of the second paragraph.
Section 2: Merger and AGN:
R: In this section describing AGN luminosity and merger as a trigger, it may be a good idea to include the following two relations from ULIRG studies available through the ARAA review paper by Sanders & Mirabel (1996). There is an inverse relationship between internuclear distance and IR luminosity of ULIRGS. Furthermore, the AGN fraction increases with the increase of IR luminosity, demonstrating merger as a trigger and possible evolution of star formation to AGN activity.
A: We agree that ULIRGs are interesting objects in the context of this discussion, and we added a paragraph on them in the section about mergers.
Section 2.1 Individual Cases.
R: Perfect examples were given for the physical process we are interested in. I would like to draw author's attention to the case of NGC3801 where multi-wavelength data show a nice evolutionary sequence of merger, wind and jet, shock and outflows (Croston et al. 2007 and Hota et al 2012).
A: We thank the referee for pointing out this source. We have added a paragraph with discussion of this galaxy.
Section 3: radio mode feedback:
R: In this section, I would suggest addition of the fact that radio mode feedback is not confined to narrow linear jet structures but also radio bubbles, specially so in sub-galactic scale. Some brilliant examples are Radio bubbles in our own Milky Way galaxy recently imaged with MeerKAT, NGC3079 (Cecil et al. 2001), Circinus Galaxy (Elmouttie et al. 1998) and a dozen or so examples compiled by Hota & Saikia 2006. Collimation of the jet is probably lost in many cases due to jet-ISM interactions and/or precession of the jet. Authors may add a couple of examples in a suitable place in this subsection.
A: We have added a short subsection on radio bubbles. Even though their origin is debatable between nuclear starbursts and AGN, as a source of radio feedback these are important to mention, and we thank the referee for pointing this out.
Subsection 3.2:
R: Along with expressing that molecular gas dominates over the ionised gas fraction in the outflow, it may be useful to express them in numbers. Outflow rates being comparable to the star formation rate is a very convincing argument for AGN feedback.
A: We have added a short text giving some numbers of mass outflow rates, comparing molecular and ionized gas phases. We also mention the typical SFR in the same AGN luminosity range.
Section 4: feedback perpendicular to the jet
R: This is not only a recent result but also seems to be the highlight of this review. Again, most studies in the past were either narrow band imaging or long-slit spectroscopy. Hence, IFU is going to lead the result in this new direction.
Instead of putting images from other sources that cover the full page of the paper, I would strongly advise for a schematic diagram highlighting this new kinematic component in a galaxy seen orthogonal to the jet.
A: Although we agree with the referee that a schematic diagram would help visualize the geometry of the outflow perpendicular to the radio jet, we believe that it may be more useful to show how this appears in observations. We believe that Fig. 7 illustrates this geometry very well, together with Fig. 8, that shows the result of simulations. We thus would argue to keep Figs. 7 and 8 as illustrations of this component.
Section 5: Red Geysers:
R: The second paragraph describes an important caveat. Dividing samples with radio-loud and radio-quiet leads to a lot of confusing correlations or lack of correlations. Radio continuum feature, jet, is a short-term phenomena and most continuum optical/IR features are orders of magnitude older. Radio may be a few tens of million years old and an azimuthally symmetric optical feature in a galaxy may be a few 100 million years old. Radio jet and dust disks/rings seen in early-type galaxies do not represent feedback from jet reaching stellar population but must be from much earlier episodes of either jet/wind feedback, after the merger (e.g. Cen A, NGC3801).
In the next paragraph, on red-geysers, authors have mentioned precession of the jets for explaining misalignment of ionised gas on a larger and smaller scales and with the radio features. I think that these radio observations are limited to detection of compact/young/high-radio frequency features but when observed with low-frequency images like LOFAR/GMRT we may detect older episodes of jet feedback, coinciding with ionised gas detections at larger-scales. To explain some of the ionised gas seen orthogonal to jets, a speculation may be added that such things could also be due to episodic nature of jets which need not be in the same orientation as the recent jet but all episodes with in the ISM or stellar light distribution of the galaxy (NGC 5813; Randall et al. 2011, Mrk6: Mingo et al. 2011, Saikia & Jamrozy 2009). In fact LOFAR is now detecting large samples of galaxy 10-80 kpc scale low luminosity radio emission which are in between the kpc-scale radio emission in Seyferts and 300 kpc scale lobes of powerful radio galaxies (Webster et al. 2020). Future SKA may add to this study. Looking forward to the modified version. with best wishes...
A: We thank the referee for these comments, which are indeed very relevant to the discussion. We have added a few sentences regarding the radio jet time scales and the new detections using LOFAR in this section.
.
Reviewer 2 Report
The article is a review on the observation of the feedback deposited locally in the central region of the host galaxies.
I am happy with how the authors have addressed this paper. In my opinion, the manuscript can be accepted for publication with minor revision.
The manuscript lacks putting the results of this study in the context of other statistical studies from literature. Zhang+2011 study of ionised outflows in z<0.8 Seyfert 1s and QSOs from SDSS. Perna+2017 for outflows in z<0.8 SDSS AGN, the outflow study of Mullaney+2013 of 24264 type 1 and 2 AGN from SDSS (see e.g. their comparison of outflow kinematics and incidence with radio loudness, Eddington ratio, AGN type). The outflow study in Wylezalek & Zakamska 2016 with 132 radio-quiet type-2 and red AGN at 0.1 < z < 1.
I will like authors include a review of redshift evolution of Quasars and Seyfert galaxies cause of this work only include nearby galaxies. I know that spatial resolution is important but there are many works on that.
Author Response
ANSWERS TO REFEREE 2
We thank the anonymous referee for the many useful suggestions, which helped us improve the manuscript. Changes to the manuscript are shown in red in the new version and answers to specific points are given below.
Referee: The article is a review on the observation of the feedback deposited locally in the central region of the host galaxies.
I am happy with how the authors have addressed this paper. In my opinion, the manuscript can be accepted for publication with minor revision.
The manuscript lacks putting the results of this study in the context of other statistical studies from literature. Zhang+2011 study of ionised outflows in z<0.8 Seyfert 1s and QSOs from SDSS. Perna+2017 for outflows in z<0.8 SDSS AGN, the outflow study of Mullaney+2013 of 24264 type 1 and 2 AGN from SDSS (see e.g. their comparison of outflow kinematics and incidence with radio loudness, Eddington ratio, AGN type). The outflow study in Wylezalek & Zakamska 2016 with 132 radio-quiet type-2 and red AGN at 0.1 < z < 1.
Authors: We have tried to focus on the radio mode feedback and its relation to the host galaxies in resolved studies, which is the scope of our study. But we thank the referee for the suggestions and we have included in the revised manuscript a new small Section (Section 4 in the revised paper) where we have included some text on the mentioned statistical studies, in particular those of Mullaney+13 and Wylezalek & Zakamska 16 papers, as they seem to fit closer to our discussion.
R: I will like authors include a review of redshift evolution of Quasars and Seyfert galaxies cause of this work only include nearby galaxies. I know that spatial resolution is important but there are many works on that.
A: This is a very interesting and important topic, but we believe that such a review could be a paper of its own and beyond the scope of our work. In order to address at least partially this request, we have included in the new Section 4 also the discussion of the work of Delvecchio 2017 that discusses the evolution of radio AGN and their hosts out to z~6.